# Genetic Variability in the E6/E7 Region of Human Papillomavirus 16 in Women from Ecuador

**DOI:** 10.3390/v15061393

**Published:** 2023-06-19

**Authors:** Alicia Zhingre, César Bedoya-Pilozo, Diana Gutiérrez-Pallo, Inés Badano, Andrés Herrera-Yela, Zoila Salazar, Damaris Alarcón, Natali Argüello-Bravo, Maylen Espinoza, Patricio Ponce, Yudira Soto, Andrés Carrazco-Montalvo

**Affiliations:** 1Laboratorio de Biotecnología, Universidad Católica de Cuenca, Cuenca 010107, Ecuador; azhingres@ucacue.edu.ec (A.Z.); zsalazart@ucacue.edu.ec (Z.S.); 2Hospital Luis Vernaza, Guayaquil 090306, Ecuador; cbedoya@jbgye.org.ec; 3Lab Genetics Ballenita, Santa Elena 241701, Ecuador; 4Instituto Nacional de Investigación en Salud Pública “Leopoldo Izquieta Pérez”, Centro de Referencia Nacional de Genómica, Secuenciación y Bioinformática, Quito 170403, Ecuador; dgutierrez@inspi.gob.ec (D.G.-P.); dalarcon@inspi.gob.ec (D.A.); 5Laboratorio de Antropología Biológica y Bioinformática Aplicada, Red de Laboratorios, Universidad Nacional de Misiones and Consejo Nacional de Investigaciones Científicas y Técnicas (CONICET), Posadas N3300, Argentina; inesbadano@gmail.com; 6Facultad de Ingeniería, Maestría en Biología Computacional, Pontificia Universidad Católica del Ecuador, Quito 170803, Ecuador; maherrerayela@hotmail.com; 7Instituto Nacional de Investigación en Salud Pública “Leopoldo Izquieta Pérez”, Laboratorio de Influenza y Otros Virus Respiratorios, Quito 170403, Ecuador; aarguello@inspi.gob.ec; 8Hospital IESS CEIBOS, Guayaquil 090615, Ecuador; maylenespinosa1@gmail.com; 9Instituto Nacional de Investigación en Salud Pública “Leopoldo Izquieta Pérez”, Gestión de Investigación, Desarrollo e Innovación, Quito 170403, Ecuador; wponce@inspi.gob.ec; 10Instituto de Medicina Tropical “Pedro Kouri”, Laboratorio de Infecciones de Transmisión Sexual, Departamento de Virología, Autopista Novia del Mediodía Km 6 ½, La Habana 10100, Cuba; yudira700618@gmail.com; 11Departamento de Biología Molecular, Centro de Biología Molecular Severo Ochoa (CSIC-UAM), Universidad Autónoma de Madrid, 28049 Madrid, Spain

**Keywords:** human papilloma virus, genetics, cervical cancer, lineage

## Abstract

Human Papillomavirus (HPV) infection is associated with intraepithelial neoplasia and cervical cancer (CC). Ecuador has a high prevalence of cervical cancer, with more than 1600 new cases diagnosed annually. This study aimed to analyze oncogenes E6 and E7 of HPV16 in samples collected from women with cancerous and precancerous cervical lesions from the Ecuadorian coast. Twenty-nine women, including six with ASCUS, three with LSIL, thirteen with HSIL, and seven with Cacu, were analyzed. The most common SNPs were E6 350G or L83V (82.6%) and E6 145T/286A/289G/335T/350G or Q14H/F78Y/L83V (17.4%). Both variants are reported to be associated with an increased risk of cervical cancer in worldwide studies. In contrast, all E7 genes have conserved amino-acid positions. Phylogenetic trees showed the circulation of the D (26.1%) and A (73.9) lineages. The frequency of D was higher than that reported in other comparable studies in Ecuador and Latin America, and may be related to the ethnic composition of the studied populations. This study contributes to the characterization of the potential risk factors for cervical carcinogenesis associated with Ecuadorian women infected with HPV16.

## 1. Introduction

Human Papillomavirus (HPV) is one of the most common genital infections worldwide and is classified as a sexually transmitted infection (STI) [1]. It is estimated that more than 80% of sexually active people will become infected with the virus at some point in their lives [2]. HPV is transmitted by skin-to-skin contact, mainly during sexual intercourse [3]. There are 448 genotypes of HPV detected, and 12 are classified as high-risk types [4,5]. Most people infected with HPV have no symptoms, so it is difficult to know if they are infected. Some people may experience genital warts or cervical changes that can be detected during a Pap test or Pap smear; however, conventional detection technologies have limitations because they may not detect the specific HPV type or genetic variant of infected cells [1,6,7].

HPV infection is associated with cervical intraepithelial neoplasia (CIN) and cervical cancer (CC). The most frequent high-risk oncogenic genotypes are HPV 16 and 18, which are linked to different types of cancers, including cervical, vulvar, and vaginal cancers in women, and cancer of the penis, anus, mouth, and throat in men [6]. According to the World Health Organization (WHO), about 530,000 new cases were reported in 2012, with a 7.5% HPV-associated female mortality. The incidence and mortality rates of CC are very high in sub-Saharan Africa, Latin America, and Southeast Asia [6]. HPV16 is classified as high-risk according to its capacity to produce cervical intraepithelial neoplasia. Persistent infection by these oncogenic genotypes leads to CC development. The genotype HPV16 is responsible for 50–60% of all cases [7]. The viral genome contains eight open reading frames (E1, E2, E4, E5, E6, E7, L1, and L2), a long non-coding control region (LCR) and a short non-coding region (NCR) located between E5 and L2 [8].

Mutations in HPV16 have been identified in genes encoding the E6 protein. This gene is involved in genomic instability of human cells through its interaction with p53, which may lead to altered carcinogenic potential and contribute to increased pathogenicity, whereas the amino acid conservation in the E7 protein is associated with cervical cancer development [9,10,11].

HPV16 mutations can become fixed in the HPV16 genome as single nucleotide polymorphism (SNPs) defining viral lineages and have been observed in different populations worldwide [12,13]. Historically, the HPV community has defined variants as isolates with less than 10% sequence diversity and described them according to their geographic location and specific changes at nucleotide positions [14]. Later, phylogenetic approaches allowed one to group them into monophyletic groups named lineages, termed A (European), B (African type 1), C (African type 2) and D (Asian American), with nucleotide sequence differences ranging from >1% to <10% and sublineages with differences ranging from <0.5% to >1% (A1–A4; B1–B4, C1–C4, D1–D4) [15,16,17].

In Ecuador, more than 1600 new cases of CC are diagnosed each year (estimated data for 2018), making it the second leading cause of cancer-related deaths among women aged 20–69 years. According to GLOBOCAN, Ecuador ranks seventh in the region, with the highest prevalence of CC, after Chile [18]. In 2014, Ecuador experienced its highest peak of deaths from this disease, which represented the leading cause of cancer-related deaths, surpassing breast cancer by 4% and stomach cancer by 0.5%. In the same year, the Society for the Fight Against Cancer (SOLCA) reported that 20 out of every 100,000 women suffered from some type of neoplasm, and CC ranked second as the leading cause in the cities of Quito and Loja, with 34.1% and 35.6%, respectively [19].

Therefore, this study aimed to analyze mutations in the HPV16 E6/E7 regions in samples from Ecuadorian coastal women with cancerous and precancerous cervical lesions.

## 2. Materials and Methods

### 2.1. Sample Collection

In this study, 29 samples from women with squamous intraepithelial lesions and cervical cancer were studied: 6 ASCUS (Atypical squamous cells of undetermined significance), 3 LSIL (low-grade squamous intraepithelial lesion), 12 HSIL (high-grade squamous intraepithelial lesion) and 7 CC (cervical cancer) attending six community health centers from the Coastal region of Ecuador between 2012–2015 were available for sequencing, including the localities of Esmeraldas (n = 4), Guayas (n = 14), Santa Elena (n = 1), El Oro (n = 1), Los Rios (n = 4), and Manabí (n = 5). All patients had previous results corresponding to HPV16 infection, as described by Bedoya-Pilozo et al., 2018 [20]. This study was approved by the Bioethics Committee (Hospital “Francisco Icaza Bustamente”, Ministry of Public Health).

### 2.2. Molecular Analysis

For amplification of the E6 gene, we used the pair of primers E6-F (5′-CGAAACCGGTTAGTATAA ‘-3′) and E6-R (5′-GTATCTCCATGCATGATT-3′), and for E7 we used the primers E7-F 5′-ATAATATAAGGGGTCGGTGG-3′ and E7-R 5′-CATTTTCGTTCTCGTCATCTG-3′R [21,22]. The primer-annealing regions were E6 (nucleotides 52-575) and E7 (nucleotides 480-985), both of which flank the coding regions of these genes (nucleotides 104-559 and 562-858, respectively). PCR reactions were performed in a final volume of 25 µL with 5 µL of DNA and 10 µM of each primer, and the cycling amplification profile conditions were as follows: five minutes at 94 °C, followed by 35 cycles of 60 s at 94 °C, 60 s at 55 °C, and 60 s at 72 °C, with a final extension at 72 °C for seven minutes [23]. The PCR amplicons were detected by electrophoresis on a 2% agarose gel in TAE buffer, stained with SYBR^®^ Safe 10,000× (Invitrogen), and purified using the PCR Purification Kit (Qiagen). The resulting PCR products of 524 pb for E6 and 506-bp for E7 were sequenced with the original primers and analyzed separately. Briefly, the purified amplicons were sent for sequencing using the Sanger method (ADN ABI 3730xl) to Genewiz, NJ, USA. The chromatograms were manually curated, cleaned and analyzed using Codon Code Aligner software (CodonCode Corporation). The sequences were run through NCBI BLAST to confirm viral origin [24].

### 2.3. Genetic Characterization and Phylogenetic Analyses

The E6 and E7 sequences were aligned, and single nucleotide polymorphisms (SNPs) were identified using Codon Code Aligner Software [Codon Code Corporation]. An isolate was classified as a variant if it had at least one nucleotide substitution change (polymorphism) when compared with the reference isolate [16].

Phylogenetic analysis of the E6 oncogene was performed using the maximum likelihood (ML) method in IQtree 2.2.0 [25]. The dataset comprises 23 sequences from Ecuador along with 10 HPV16 reference genomes: A1 (European, E) K02718; A2 (E) AF536179; A3 (E) HQ644236, A4 (Asian, East) AF534061; B1 (African type-1a, Afr1a) AF536180; B2 (African type-1b, Afr1b) HQ644298; C (African-2, Afr2a) AF472509; D1 (North America, NA) HQ644257; D2 (Asian American, AA2) AY686579; D3 (AA) AF402678 [15]. The dataset was aligned using MAFFT, and the phylogenetic tree was estimated using the Tamura 3-parameter substitution model and 1000 bootstrap replicates [26].

### 2.4. GenBank Accession Numbers

The sequences described in this study were deposited in GenBank under the following accession numbers: E6: OQ730038–OQ730060 and E7 OQ730061–OQ730081.

## 3. Results

### 3.1. Sample Characteristics

The study sample comprises 29 women with cervical lesions, including 6 ASCUS, 3 LSIL, 13 HSIL and 7 CC. The median age was 49.5 (age range: 30–76 years).

### 3.2. E6 and E7 Genetic Characterization

Sequence data for the E6 gene were obtained from 23 samples (23/29, 79.3%), and we identified four different variants. The most common was E6 350G or L83V (82.6%, 19/23), and E6 145T/286A/289G/335T/350G or Q14H/F78Y/L83V (17.4%, 4/23). The E7 gene was sequenced in 21 samples (72.4%). The most common variant was the prototype, followed by E7 732C/789C/795C (23.8%, 5/21). All SNPs in E7 were silent mutations. Details of the SNPs and lineages are shown in Table 1. Both genes are described independently. The HPV lineages do not add up for both genes since there was some information missing at the involved genes (undetermined by sequencing).

### 3.3. Phylogenetic Analysis

The phylogenetic classification of E6 sequences is shown in Figure 1. The tree topology retrieves the four major lineages of HPV16 evolution: A (European), B and C (African), and D (Asian American). Samples from Ecuador corresponding to Guayas (n = 10), Esmeraldas (n = 2), Manabí (n = 1), Santa Elena (n = 1), and Los Ríos (n = 3) were assigned to lineage A; and samples corresponding to Guayas (n = 3), Esmeraldas (n = 2), and Manabí (n = 1) were assigned to Lineage D. There were no samples assigned to lineages C and D (African origin).

## 4. Discussion

Human papillomavirus (HPV) is responsible for 4.5% of all human cancers, with CC being the most common. In 2020, CC became the fourth most frequent cancer among women worldwide, with approximately 342,000 new cases diagnosed annually in the world [27]. A total of 222 genotypes of the virus have been detected in humans [28], of which 12 are classified as carcinogenic genotypes: 16, 18, 31, 33, 35, 39, 45, 51, 52, 56, 58 and 59 [5]. The development of this type of cancer and the appearance of precancerous lesions are directly related to infection by high-risk HPV, particularly by the oncogenic genotypes HPV16 and HPV18 [5,7].

HPV genotype 16 is most commonly found in cervical precursor lesions and cervical cancer, with an odds ratio association value of over 300 [5]. Additionally, several epidemiological studies have shown that non-European variants of HPV16 (lineages B/C/D) have a stronger association with high-grade cervical neoplasia and cancer than the European lineage (A) [12,13,16]. Mutations in the E6 and E7 genes may influence the processes of malignant transformation [10]. The E6 gene is located between nucleotides 104 and 559, whereas E7 is located between nucleotides 562 and 858 [29]. Mutations in the E6 gene are associated with cell cycle arrest and the absence of apoptosis, whereas the strict conservation of the E7 gene is associated with cervical cancer cells [10,11].

In this study, the most common variant was E6 350G or L83V (82.6%, 19/23). From an evolutionary perspective, the E6 350G SNP has arisen independently in different HPV16 lineages [12,13]. In this study, we found the 350G SNP in both lineage A (56.5%) and D samples (26.1%), consistent with previous reports [13]. Interestingly, in vitro studies have shown that keratinocytes infected with E6 350G have a higher capacity for cell transformation compared to those infected with E6 350T, regardless of their evolutionary origin [13], suggesting that these SNPs are potential molecular markers of cancer progression [10]. Another SNP found in E6 gene was the A532G polymorphism (six samples). This is a synonymous substitution that does not change the amino acid sequence of the E6 oncoprotein and was previously identified in HPV16 samples collected in Korea [30]. However, its biological meaning is unknown.

Regarding the E7 oncogene, all identified SNPs were synonymous substitutions. This strict conservation of the 98 amino acids of E7 (which disrupts Rb function) is critical for HPV16 carcinogenesis and has been indicated as a risk factor in large worldwide studies [11]. Briefly, we identified C678T and T749C polymorphisms in one sample. The C678T mutation has been previously reported by Antaño et al. in 0.53% of the samples from Mexico [31]. In addition, polymorphisms C732T, C789T, and G795T were identified in five samples. These SNPs are also synonymous mutations and are usually found linked, resulting in a pattern indicative of lineage D [12]. Therefore, they have been associated with an increased risk of CC [12,13]. In a study by Antaño et al. on variants of the E6 and E7 genes of HPV16 in women from southern Mexico, the combination of the three polymorphisms found (E7-C732/C789/G795) was associated with a 3.79 increased risk of developing CC compared to the wild-type genotypes of the E7 gene [31], a result probably attributable to lineage D. Furthermore, HPV16 variants from lineage D have been associated with invasive CC in other Latin American countries [32].

Finally, phylogenetic analyses based on the E6 region identified that the Ecuador sequences belonged to lineages A and D. These lineages have different origins, and are differentially distributed worldwide. For example, lineage A is frequently found in North America and Europe, whereas lineage D is more frequently found in Asia and America [14,16]. In this study, 26.1% of the samples belonged to lineage D. This result is in contrast to the frequency reported for Quito, Ecuador by Mejía et al. (6.6%) [33]. The higher prevalence of HPV16 lineage D may be related to factors such as the ethnic composition of the population. This suggests the possibility that the distribution of this lineage in the littoral region may be related to historical links and migrations within the country, and it deserves future studies including human genetic markers [34].

Other studies of comparable size have also shown a different prevalence of lineage D compared to this study. For example, a study involving 38 cervical lesions from Mexico found a frequency of lineage D of 5.3% (2/38) [35]. Another study from Mexico involving 20 healthy women and 21 cervical lesions/cancer cases (considering only the case groups for comparison) found a frequency of lineage D of 9.5% [36]. A study from Brazil involving 20 cases of cervical lesions (LSIL and HSIL) in HIV-negative women found a prevalence of lineage D of 15% [37]. Finally, a larger case-control study from Argentina by Totaro et al. (2021) identified a frequency of lineage D of 6% among 83 samples of L-SIL/HSIL/cancer lesions [38]. Nevertheless, future studies with large sample sizes will help elucidate the epidemiology of HPV16 lineages in this population. Another limitation of our study is that the E6 gene was not phylogenetically informative for identifying sublineages or for recovering the reference tree topology described by whole genome analysis, in which lineages C and D share a common ancestry [15]. However, the sequence was sufficient to recover supported phylogenetic clusters for the classification of lineages A and D. In addition, we identified co-infection of lineages A and D in four samples. Recent studies using Next-Generation Sequencing (NGS) have shown that co-infections are surprisingly common [39], highlighting the importance of addressing this issue with additional methodologies in future research. Finally, the lack of successful sequencing in nearly 20% of the samples may be attributed to low sample quality and/or a low viral load in the lesions analyzed, among other possible factors.

This study contributes to the genetic characterization of the E6 and E7 HPV oncogenes in samples from Ecuadorian women. The identification of well-described molecular markers of cancer progression, such as the conservation of E7 protein, HPV16 lineage D, and E6 SNPs 350G, will help identify women at an increased risk of developing CC.

## Figures and Tables

**Figure 1 viruses-15-01393-f001:**
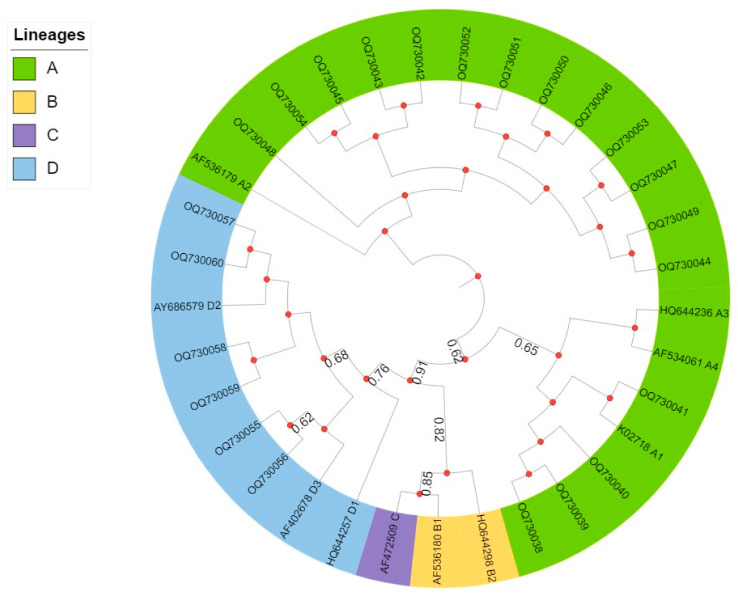
Phylogenetic tree of HPV16 based on E6 oncogene corresponding to 23 sequences of analyzed samples. Each lineage is represented by a different color: lineage A (lime green), lineage B (mustard), lineage C (purple), and lineage D (light blue). Bootstrap values greater than or equal to 50 are represented by circles. Maximum Likelihood method, Tamura 3-parameter substitution model, 1000 bootstrap replicates.

**Table 1 viruses-15-01393-t001:** HPV16 polymorphisms and lineages identified at E6 and E7 genes.

E6	145	183	286	289	335	350	532	Amino Acids	n (%)	Cervical Lesions				Lineage
Genome Position														
Ref SNPs	G	T	T	A	C	T	A	Ref		ASCUS	LSIL	HSIL	CC	
OQ730038	-	-	-	-	-	-	-	Ref	4 (17.4)	1	1	1	1	A
OQ730042	-	-	-	-	-	G	-	L83V	13 (56.5)	2	-	8	2	A
OQ730057	T	-	A	G	T	G	G	Q14H–H78Y–L83V	4 (17.4)	1	-	3	-	D
OQ730055	T	G	A	G	T	G	G	Q14H–I27R–H78Y–L83V	2 (8.7)	-	-	-	2	D
Total									23 (100)	5	1	12	5	
									29	1	2	1	1	
**E7**	**678**	**732**	**789**	**795**	**828**			**Amino Acids**	**n (%)**	**Cervical Lesions**				**Lineage**
**Genome Position**														
Ref	T	T	T	T	T			Ref		ASCUS	LSIL	HSIL	CC	
OQ730063	-	-	-	-	-			Ref	13 (61.9)	2	2	4	5	A
OQ730062	C	-	-	-	-			Ref	2 (9.5)	1	-	1	-	A
OQ730076	-	-	C	-	C			Ref	1 (4.8)	1	-	-	-	A
OQ730077	-	C	C	G	-			Ref	5 (23.8)	2	1	2	-	D
Total									21 (100)	6	3	7	5	
									29	0	0	5	2	

## Data Availability

Not applicable.

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
