# Peer review of "Genetic Variability in the E6/E7 Region of Human Papillomavirus 16 in Women from Ecuador"

_viruses, 2023, doi:10.3390/v15061393_

Round 1

Reviewer 1 Report

RECOMMENDATION:

Major revision

SUMMARY:

Zhingre et al. provide a descriptive study of HPV16 genetic variation in the E6 and E7 ORFs among 29 cervical cancer samples from Ecuador. I think Viruses is a good fit. However, revisions are necessary. As currently written, the study promotes some confusion and perhaps unwittingly implies some false ideas, including (1) the idea that mutations in HPV16 E7 unambiguously promote carcinogenesis, which is contradicted by their own study; and (2) that the lineages of HPV16 are paraphyletic. Further, there is lack of clarity about the meaning of some terms, including STD and variant, inappropriate citations, and failure to explore some known sources of artifactual results. I believe all these issues can be easily remedied by more careful editing, description of methods, and very simple additional analyses of the data already in hand. These issues must absolutely be addressed before publication will be appropriate.

MAJOR POINTS:

1. The authors state that “mutations in the E7 protein are associated with cancer cell proliferation” in HPV16 (line 69), but their own results contradict this (i.e., there are no amino acid variants found in E7 for cancers), as do some studies they cite (e.g., ref [31], Mirabello et al. 2017; but see ref. [25] Antaño-Arias et al.). Please explain.

2. There are numerous cases of what appear to be the wrong references cited. For example, ref. [13] on line 64; ref. [24] on line 167; and perhaps all ref. [11] citations on lines 169-173. For instance, ref. [11] is certainly not a primary study demonstrating that “mutations in the E7 gene are associated with the hyperproliferation of cancer cells” (lines 172-173)? Please make sure the appropriate primary studies or topical reviews are cited in these and all other instances.

3. Where was the “amino acid signature for Lineage D” (lines 135, 175-176, 189) obtained? Please explain and cite the source.

4. There is lack of clarity and failure to use precisely defined terms throughout. For example, what is the meaning of ‘Concatenated genes’ on line 138? Another example: what is the meaning of ‘variants’ on line 134, where it might actually be referring to lineages?

5. For production: Table 1 is nearly impossible to read, likely the fault of production. The production team should ensure all rows and columns align well, and that there is enough horizontal space in each column for the values to be easily read.

6. Table 1: the authors should provide unique IDs of samples for both E6 and E7 so that the reader can determine which E6 variants are linked to which E7 variants. If NO variants are linked in the same sample(s), then feel free to omit this suggestion, but please state it explicitly.

7. The authors must check whether any variants occur within primer regions. If so, make sure to discuss any possible implications for sequencing biases.

8. What steps, if any, were taken to rule out coinfection? Coinfection with different sub/lineages of HPV16 is surprisingly common, given that this is the most prevalent carcinogenic HPV type. It is possible that some of the variants, particularly those at lineage-defining sites, are due to coinfection. Please provide information or analysis regarding this possibility.

9. Figure 2: this phylogeny contradicts the known HPV16 phylogeny in many of its groupings, e.g., the tree shows (A2,((A4,A3),A1)) and (((C,B1),B2),D) but the real relationships are thought to be (((A1,A2),A3),A4) and (B,(C,D)), respectively (see ref. [4], Nelson & Mirabello 2023). Importantly, this doesn’t mean there is a problem with the analysis, because it is well known that different ORFs yield different topologies (García-Vallvé et al. 2005, Papillomaviruses: different genes have different histories. Trends in Microbiology. 13, 514–521). Further, it is known that the whole genome sequence is necessary for classification of lineages and sublineages (Burk et al. 2013, Human papillomavirus genome variants. Virology. 445, 232–243). The fact that this tree topology is not the true historical phylogeny, and that these lineage assignments are somewhat ambiguous, must be carefully considered and explained in the manuscript.

10. Figure 2: the caption says that only bootstraps ≥50% are shown, but according to the sizes shown in the key, there is nothing shown less than ~70%. Please check if the key is correct, and consider giving it sensible cutoffs, e.g., 50 / 75 / 80 / 90 / 95.

11. Line 206 paragraph: this interpretation may be mistaken. The present study is explicitly one of cancers, but at least some of the citations appear to be samples from screening of the general population (both healthy samples AND cancers). Because it is known that D lineage genotypes are the most carcinogenic (Mirabello et al. 2016, HPV16 Sublineage Associations With Histology-Specific Cancer Risk Using HPV Whole-Genome Sequences in 3200 Women. Journal of the National Cancer Institute. 108, djw100), their enrichment in cancer samples is not necessarily surprising. Please edit this section to ensure a ‘fair comparison’ with ONLY studies of cancer samples (feel free to note why this might differ from screening studies), and revise interpretation if necessary.

12. Throughout, L83V is termed a lineage D ‘diagnostic SNP’. The nucleotide change underlying this is T350G. However, although T is approximately fixed in lineages B and C, large-scale sequencing studies show lineage A is approximately 50% T, 50% G. Please make it clear that mere presence of the L83V (T350G) allele is also perfectly consistent with lineage A virus, where the variant is also prevalent, and adjust any interpretation if necessary.

ADDITIONAL POINTS:

1. HPV is a sexually transmitted infection (STI), not a sexually transmitted disease (STD) (line 46). An STI may or may not give rise to disease, but the virus itself is not a disease. Please modify accordingly.

2. What is meant by “neoplasia at the epithelial level”? (line 62)

3. The genome also consists of a short non-coding region (NCR) falling between E5 and L2 (line 65)

4. Figure 1 should, at the very least, have some description that explains how the map was made and the significance of what is being shown. All decimals and significant figures (.000) are unnecessary and should be eliminated. Label the axes.

5. Line 160: cite primary source for number of deaths.

6. Lines 167-168: as currently worded, the sentences risks implying that some genotypes of HPV16 might NOT have carcinogenic potential (ability to cause cancer), which is almost certainly false in general. Change to something like, “The risk of cancer… changes depending on…”.

PROOFREADING:

Line 71: ‘lineages’ (lowercase)

Line 99: “E7-F” and “E7-R”

Line 146: ‘Asian American’

Line 147: ‘assigned’

Line 149: ‘assigned’

Line 150: ‘African’

Line 162: replace ‘y’ with ‘and’

Line 164: replace ‘specifically’ with ‘particularly’ (cancer is not only due to these two genotypes!)

Line 166: “identified in cervical precursor lesions”

Line 200: ‘26.1%’

Quite good, minor editing required.

Author Response

Dear,

Please see the attachment. Thank you for your suggestions.

Reviewer 2 Report

The paper entitled ” Genetic Variability in the E6/E7 Region of Human Papillomavirus 16 in Women from Ecuador“ by Alicia Zhingre et al., was designed to analyze HPV16 E6 and E7 oncogenes in 29 samples collected from women from the Ecuadorian coast  with precancerous and cancerous cervical lesions. Specimens corresponded to 6 ASCUS, 3 LSIL, 13 HSIL, and 7 CC. The most common SNPs found in E6 were 350G or L83V (82.6%) whereas all SNPs in E7 corresponded to silent mutations.

This study is an interesting contribution to the characterization of HPV16 linage present in Ecuador. The author underline that the results will help to identify women at an increased risk of developing CC. Since the study included intraepithelial neoplasia and invasive carcinoma, and although there is a limited number of cases analyzed in each category, it would be interesting to know if the repartition of the variants was different according to the type of lesion (ASCUS/Low grade VS High grade/Ca).

Minor remark: y line 162 

English is correct

Author Response

(The authors gave the same response as above.)

Reviewer 3 Report

In the manuscript by Zhingre et al the authors describe a small study of genetic variability of HPV16 in Ecuador. The study is mostly well written but suffers from several issues. Foremost is that the study actually assessed very few cases in total n=23 with unusually high 20% sequence failure rate indicating methodological issues from the study onset. No significant effort was put into explaining the low sample numbers or technical issues.

The fact that there are 12 authors and 23 samples is also eyebrow raising. Some parts of the study are also included with the sole reason of justifying the authorship, ie fine grained geographical location, which unfortunately doesn’t otherwise provide scientific or clinical insights.

Apparently E6 and E7 were not analyzed in combination.

Some of the issues are listed below in a page line format

P2 L49 “There are 448 genotypes of HPV detected” is still factually incorrect. Indeed there are 448 humanpapillomavirus sequences deposited at PaVE website, however, only 227 are recognized by the International Human Papillomavirus Reference Center with the other sequences being only putative novel types https://pave.niaid.nih.gov/explore/reference_genomes/human_genomes.

It might be better to acknowledge this distinction in the current manuscript to avoid confusion

P2 L55 abbreviation IEN is not really that common in other publications. Much more common is cervical intraepithelial neoplasia – CIN. Furthermore since the abbreviation IEN is not used in the rest of the manuscript it would be best to either remove it completely or replace it with more commonly used one

P2 L88  The number of HPV16 samples was surprisingly low considering the patients were collected at “several …health centers” (apparently 6 as seen at L128) between 2012-2015. More detailed study population description is warranted to explain this.

P3 L107 were chromatograms evaluated manually as well to resolve ambiguities or validate SNPs?

P3 L115 source of referent sequences should be provided as reference at this point.

P4 L133 why were the sequences obtained for only 23 / 29 samples? 20% failure is a bit unusual. Considering 6 samples were inadequate and that some sites shown on Figure 1 had very few or only a single case, it is likely that some locales actually didn’t meaningfully contribute to the study. There doesn’t seem to be adequate justification for including geographic location data since such low numbers cannot be meaningfully interpreted.

P4 L 138 what does “concatenated genes” refer to?

P6 L143 why are E7 results not shown as a tree at least as a supplement? Did lineage assignments of E6 and E7 from the same sample match?

Typos

L65 “(LCR). [8]”

L69 “proliferation. [9,10,11].”

L162 “58 y 59”

Language

L71 “observed in different populations recorded worldwide.“   recorded is obsolete

L166 “HPV genotype 16 is the most frequently identified cervical precursor lesion and CC”

Typos

L65 “(LCR). [8]”

L69 “proliferation. [9,10,11].”

L162 “58 y 59”

Language

L71 “observed in different populations recorded worldwide.“   recorded is obsolete

L166 “HPV genotype 16 is the most frequently identified cervical precursor lesion and CC”

Author Response

(The authors gave the same response as above.)

Round 2

Reviewer 1 Report

Issues have been clarified to my satisfaction. However, the new additions have errors, e.g., 'aminoacid' instead of 'amino acid', punctuation etc. Be sure to check throughout. Lineages include D1-4, not just D1-3.

Issues have been clarified to my satisfaction. However, the new additions have errors, e.g., 'aminoacid' instead of 'amino acid', punctuation etc. Be sure to check throughout. Lineages include D1-4, not just D1-3.

Reviewer 3 Report

The revised manuscript by Zhingre et al addresses most of the points raised to some extent. The materials and methods section was somewhat improved, results streamlined and the discussion notably strengthened.

The authors state in their reply that “The chromatograms were evaluated manually on Codon Code Aligner to resolve ambiguities and validate SNPs.” However the methods text still says “The chromatograms were cleaned and analyzed using Codon Code Aligner software”. The name of the software doesn’t automatically suggest it allows manual investigation and might also not be familiar to many readers. Thus to remove ambiguities it would be better to explicitly state this "manual curation" part in the manuscript to strengthen the validity of the results shown

The phylogenetic tree for E7 was provided for review purposes but unfortunately the labels are misaligned and difficult to examine. The same could be provided as a supplement figure (thus not within the manuscript itself, yet available for others as well).

Apparently, it remains impossible to increase the sample size which is still a major factor for the study robustness. The authors at least strenghtened the discussion to acknowledge this point.

The authors still do not mention whether E6 and E7 lineage identification matched (as expected) or were there some mismatches that would be interesting/concerning? Apparently from Table 1 it appears that 5 cervical cancer cases had both A and D lineages in E6 while all 5 CC cases belonged to A lineage on E7 sequencing? How common is it that a variant has Q14H – I27R- H78Y - L83V mutations in E6 and has completely referent E7 sequence? Or were different E6 and different E7 cases evaluated for CC due to sample failure?

The revised discussion briefly mentions P3L258 that there was co infection of A and D lineages in 4 cases. This is not shown in results? How was this co-infection identified? Is this the reason for Table 1 inconsistency?

The discussion mentions (L229-L231) the increased risk for E7 D lineage, yet the authors do not highlight their  opposing results where no CC case (0/5) and only 2/7 HSIL cases had E7 D lineage variants?
